# Addressing hurdles in cultured meat by exploring reduced myogenesis after bovine myoblast expansion
Maria Olenic [1], Aline Baekelandt[1], Charlot Philips [1], Florian Weiland [2] & Lieven Thorrez [1]✉

When considering non-genetically modified primary bovine cells for large-scale cultured meat production, one must account for their biological limitations, such as proliferation and differentiation capacity. This study tests these limitations in primary cells from fetal and adult bovine donors. Bovine myoblasts are expanded past 30 doublings and characterized for CD56 expression, senescence, and myogenic differentiation capacity. Contrary to the Hayflick limit, several cell lines from adult cows surpass 60 population doublings, all with normal karyotypes. One line maintains high CD56 expression for the first 25 doublings. However, differentiation capacity declines, with fetal-derived cells showing high fusion indexes early on, but minimal fusion in both adult- and fetal-derived cells past 25 doublings. Differential transcriptomic and proteomic analyses of adult myoblasts with higher versus lower fusion indices identify many significantly affected genes and pathways. Genes related to myogenesis, DNA repair, and calcium signaling, among others, are downregulated in low fusing cells. This research shows the potential to expand unmodified primary bovine myoblasts for industrial cultured meat production, but further research is needed to address the lack of differentiation in expanded cells to replicate the fibrous texture and protein composition of meat.

Since the first academic publication featuring cultured meat (CM) research in 2002[1], the field of cellular agriculture grew and diversified into many directions in attempts to solve many challenges of CM production[2,3]. Often the focus is on upscaling technologies for cell expansion[4,5], consumer acceptance[6,7] and regulatory framework[8]. And while cell lines are also developed, their expansion potential and ability to form myotubes after expansion are rarely pointed out.

Most conceptual schemes of CM production start with the harvest of a muscle biopsy from a living animal[8–10]. This is followed by isolation of primary satellite cells, which, when proliferating, give rise to their myogenic progeny—myoblasts. The myoblasts are expected to proliferate and eventually fuse into myotubes, which, upon maturation form protein-rich myofibers—the major component of CM. Myoblasts are the most obvious cell type to explore for CM production because they are the natural precursor cells to myofibers[11]. It is well known, however, that adult progenitors, including myoblasts, have limited in vitro proliferation and differentiation potential[12–14].

To estimate the number of population doublings needed from myoblasts for a substantial CM production, one needs to know the starting cell number obtained from a muscle biopsy and the number of cells that make up the desired weight of the CM batch. According to some estimates, $2.9 \times 10^{11}$ cells are required for the production 1 kg of wet cell mass[4]. The first cultured burger, however, was produced using a lower cell density of $1 \times 10^{11}$ cells per 1 kg[15]. In beef production, a typical carcass yield of a Belgian Blue bull is around 500 kg[16]. Taking these quantities into account and assuming a starting cell number of $5 \times 10^4$ myoblasts from a single biopsy, it would be necessary for myoblasts to undergo ~30 doublings (equivalent to roughly $5 \times 10^{13}$ cells) to produce a cell mass equivalent to the carcass mass of one bull (Fig. 1A). There are limited reports exploring the doubling capacity of bovine myoblasts until that point[17]. Most studies report cell doublings in the range of 10 to 25[18–22], but the absolute limit of these cells is often not thoroughly investigated.

The reduction of differentiation capacity in myoblasts, unfortunately, precedes replicative senescence. This means that even if the cells are able to reach high doubling numbers, they may lose the ability to form myotubes and accumulate sarcomeric proteins that comprise the bulk of mature myofibers[23]. The myoblast differentiation capacity, further referred to as fusion index, is usually reported as a proportion of cells able to fuse into multinucleated myosin heavy chain (MyHC), tropomyosin, or desmin positive myotubes. Myoblasts from Belgian Blue cattle were reported to have

[1]Tissue Engineering Lab, Department of Development and Regeneration, KU Leuven campus Kulak, Kortrijk, Belgium. [2]Department of Microbial and Molecular Systems (MS), Laboratory of Enzyme, Fermentation and Brewing Technology (EFBT), Campus Rabot, Ghent, Belgium. ✉e-mail: lieven.thorrez@kuleuven.be

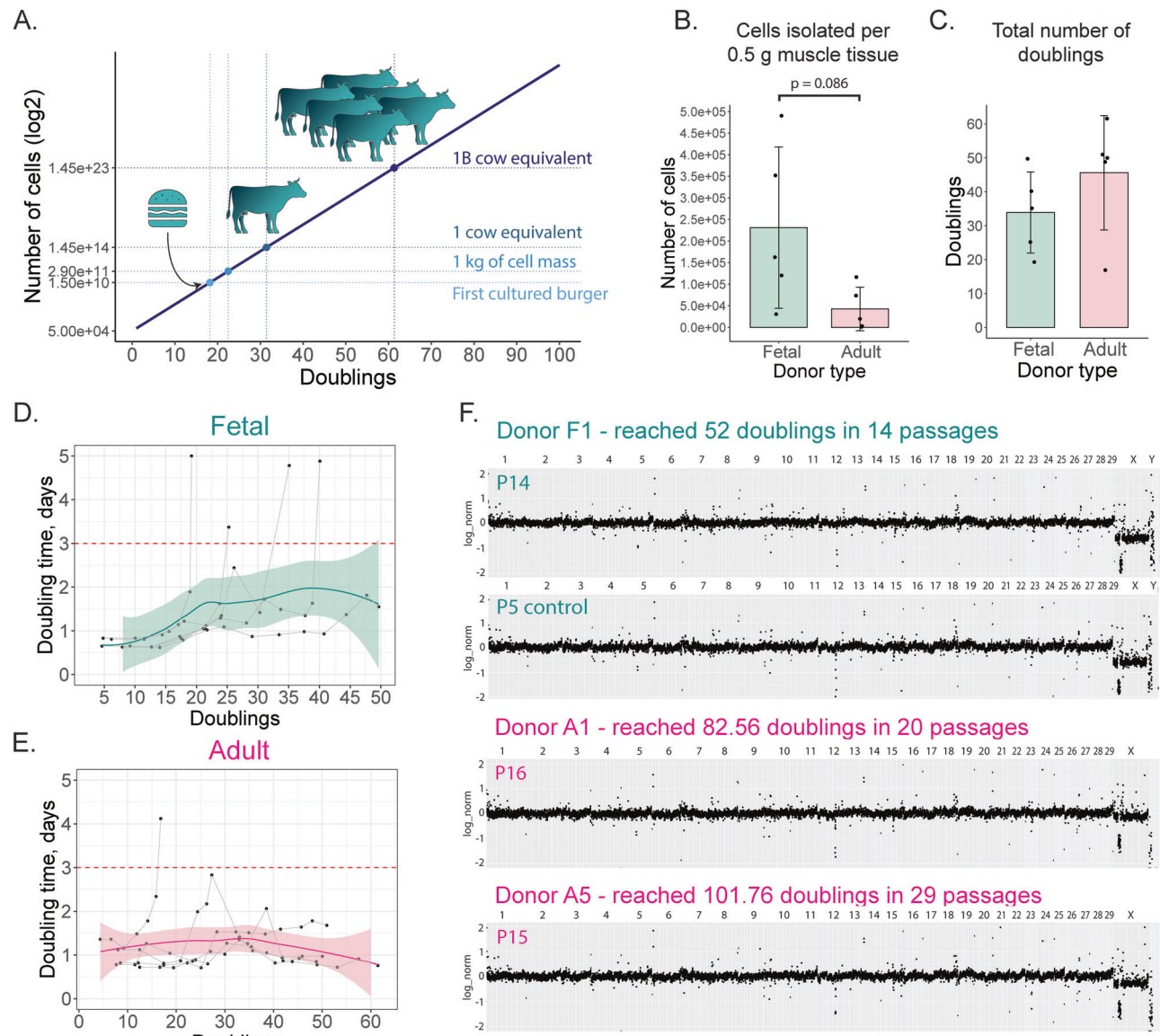

**Fig. 1 | Proliferation capacity of fetal and adult bovine myoblasts. A** Exponential function $y = 50,000*2^x$ represents the number of cells corresponding to their doubling number, assuming a starting cell number of $5 \times 10^4$ cells. Selected points representing the number of cells necessary for the production of a burger[15] or 1 kg of meat[4] are based on literature. **B** Estimation of the number of cells isolated from a 0.5 g skeletal muscle biopsy ($n = 5$, unpaired t-test). Bars are plotted as mean ± SD. **C** Total number of doublings recorded in expanded bovine myoblasts before reaching the doubling time of 3 days or 13 passages. Bars are plotted as mean ± SD. **D**, **E** Relationship between doubling time and doubling number for individual cultures of fetal and adult myoblasts. Colored lines and confidence intervals represent the loess fitted models. **F** Shallow genome sequencing of three expanded cultures with exceptional proliferation capacity shows normal karyotype.

a fusion index of ~55% at 14 population doublings, and less than 10% at 25 doublings[24]. The drastic reduction in fusion capacity was also shown in expanded myoblasts from Simmental cattle[22]. There are multiple possible contributors to this problem, including suboptimal culture conditions that induce premature differentiation[20], overgrowth of non-myogenic cells[21] and onset of early senescence, but the loss of myoblast fusion capacity was not yet characterized in detail.

Culture conditions used for myoblast expansion and differentiation are suspected to contribute to the decrease in fusion capacity the most. Historically, expansion medium with high serum content (10–30%) is being used to maintain myoblasts in proliferative state[25], and to induce differentiation, this medium is switched to the one with low serum content (e.g. 2% horse serum)[25]. Fetal bovine serum (FBS) contains mitogenic growth factors, such as bFGF and TGFβ, that promote cell division and prevent differentiation, making it a simple regulator of expansion and differentiation. In case of culturing myoblasts from Belgian Blue cattle, which have a

mutation in myostatin gene, additional co-supplementation with bFGF and p38 MAPK inhibitor[20], a downstream regulator of TGFβ pathway, is necessary to prevent premature differentiation[26]. As Belgian Blue myoblasts were used in this work, we applied this co-supplemented medium for our cells as well and compared it to the non-supplemented medium. Furthermore, the field of cellular agriculture strives to avoid animal-derived components, which restricts the use of serum as the regulator of expansion and differentiation of myoblasts. In attempt to partially substitute serum in myoblast cultures, we tried Ultroser, a known serum substitution. Some fully serum-free media, such as E8, can support short-term growth of bovine myoblasts, but cannot match serum-containing medium in performance[27]. Therefore, we explored supplementing E8 medium with albumin, an essential protein responsible for transport and buffering of many vitamins and minerals. Additionally, we explored the use of Laminin521 coating to improve cell attachment and retention during the culture in E8, as Laminin521 promotes better differentiation of human and murine myoblasts[28].

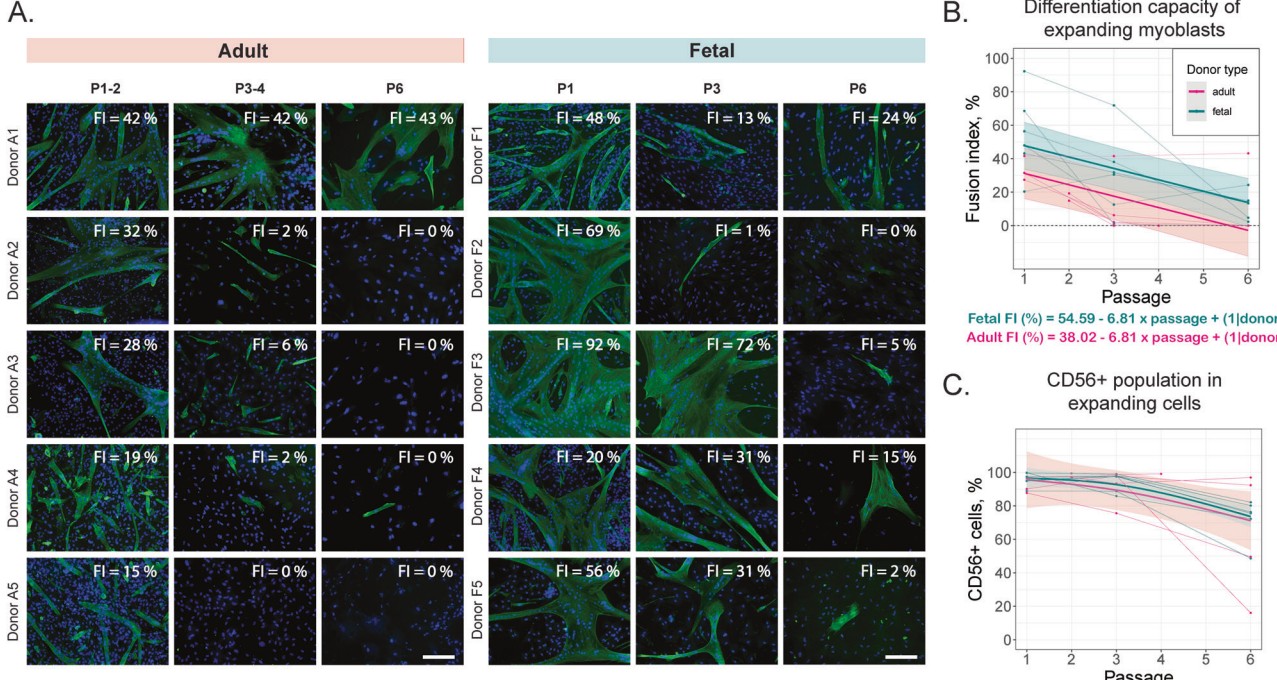

**Fig. 2 | Characterization of fusion capacity loss in expanding fetal and adult bovine myoblasts. A** Immunofluorescent imaging of tropomyosin stained myotubes (in green) and cell nuclei (in blue). A representative image from a set of 6 is shown for each donor-timepoint combination. Scale bar = 200 μm. **B** Quantitative analysis of fusion index. Images were quantified using MyoFInDer software[63]. Grey lines represent individual cultures, colored lines represent linear mixed models plotted for fetal and adult data. **C** Proportion of CD56+ cells in myoblast cultures during expansion. Colored lines represent the loess fitted models. FI fusion index, P passage.

Differentiation medium composition can also play an important role in fusion capacity determination and it is easier to come across a serum-free formulation, as it traditionally contained only a small amount of serum. We tried several published formulations[22,24,29] as well as explored the use of ERK inhibitor, which reportedly improves myoblast differentiation[30].

The age of the donor animal can also be a significant contributor to the properties of the isolated myoblasts[11]. This is a well-acknowledged concern in regenerative medicine, where cell therapy relies on the quality of the progenitors used. Quality of primary cells derived from older donors is of special concern in regenerative medicine. This is not yet studied in detail in bovine myoblasts, with only three studies thus far showing better differentiation of cells obtained from younger animals[31–33].

In the present work, we first selected expansion and differentiation conditions that allowed substantial expansion and better differentiation of myoblasts from Belgian Blue cattle. Then we characterized the proliferation and differentiation limits of myoblasts from two sources: fetal and adult skeletal muscle of Belgian Blue cattle. Myoblasts from several fetal and adult donors expand beyond 30 population doublings, but were not able to retain their fusion capacity after expansion. Upon further testing, it was confirmed that the reduction of differentiation capacity in the first passages is not caused by the overgrowth of fibroadipogenic progenitors nor the manifestation of senescence. We then performed a transcriptomic and proteomic analyses of myoblasts with higher versus lower fusion indices (HFI and LFI, respectively) and identified possible pathways and mechanisms underlying the difference in fusion capacity.

## Results
### Proliferation capacity of fetal and adult bovine myoblasts
To determine the total expansion capacity of primary bovine myoblasts, we performed isolations from fetal and adult skeletal muscle biopsy samples (*n* = 5 per group), and expanded them up to 13 passages or until their population doubling time exceeded 3 days (Supplementary Data 1). To select the most suitable expansion medium for long-term bovine myoblast

culture, we first tested different media formulations (Supplementary Table 2, Supplementary Data 2). The results showed that PM+ medium achieved sufficient expansion, prevented premature differentiation, preserved better cell morphology, and appeared to retain higher differentiation capacity in myoblasts of most donors (Supplementary Figs. 2 and 3, Supplementary Data 2). Therefore, PM+ medium was chosen for subsequent experiments.

We estimated the number of isolated myoblasts by using the number of cells obtained at first passage and the doubling time between P1 and P2. From 0.5 g of adult muscle tissue, on average $4 \pm 5 \times 10^4$ cells were obtained, while from 0.5 g of fetal tissue, $23 \pm 19 \times 10^4$ cells were obtained (unpaired t-test, p = 0.086) (Fig. 1B, Supplementary Data 1). The average total number of doublings recorded in this experiment was 34 for fetal cells and 46 for adult, with no significant difference between the two groups (*p* = 0.24) (Fig. 1C–E). Myoblasts from two adult donors and one fetal donor were expanded beyond 13 passages to determine the maximum number of doublings that can be reached before the doubling time exceeded 3 days. Adult cells reached 83 and 102 doublings, and fetal 52. The expanded cells had a normal karyotype, with no major chromosomal aberrations (Fig. 1F). A deletion in the X chromosome was present in all samples (including control cells from an early passage), and is the result of a difference between the Belgian Blue breed and the breed of the reference genome cow (Hereford, female).

### Fusion capacity of expanding fetal and adult bovine myoblasts
The differentiation capacity was monitored in expanding adult and fetal bovine myoblasts between P1, P3, and P6 (*n* = 5) (Fig. 2A, Supplementary Data 1). To select the most suitable differentiation medium, different formulations previously reported in literature were again tested (Supplementary Table 3, Supplementary Data 2). Since NL15 medium and SFDM resulted in inferior myotube formation, and the addition of ERKi did not significantly improve the differentiation, it was chosen to move forward

with the SkFM medium which formed larger, interconnected myotubes (Supplementary Fig. 4, Supplementary Data 2).

The fetal and adult myoblast cell populations had reached 6.6 ± 1.6 and 9.6 ± 2.6 doublings at P1, 16.1 ± 1.6 and 15.5 ± 4.0 doublings at P3, and 24.4 ± 2.7 and 28.0 ± 1.8 doublings at P6, respectively. A linear mixed model (with passage and donor type as fixed effects and individual donor as a random effect) was fitted to describe the temporal changes in the fusion index. The model's intercept, corresponding to adult myoblasts at P0 is at a fusion index of 38.02% (95% CI [21.88, 54.15], $p < 0.001$). Fusion index reduced by 6.81% after each passage (95% CI [−9.81, −3.81], $p < 0.001$). Fetal donors had on average a higher fusion index than adult (by 16.57%), but this effect was not statistically significant due to the limited test power ($p = 0.057$). The models for each donor type are summarized in Formulas 1 and 2 (F1, F2) and Fig. 2B.

$$\text{F1} \qquad Fetal\ Fusion\ Index\ (\%) = 54.59 - 6.81 * passage + (1|donor)$$

$$\text{F2} \qquad Adult\ Fusion\ Index\ (\%) = 38.02 - 6.81 * passage + (1|donor)$$

### Proportion of CD56⁺ population in expanding cells

To identify whether the loss of the fusion capacity of the expanding myoblast cell population was related to the overgrowth of contaminating cell populations, such as CD56⁻ fibroadipogenic progenitors, an analysis of CD56⁺ cells was performed using flow cytometry ($n = 5$). No significant decrease in CD56⁺ population was observed between P1 and P3 with the average percent of CD56⁺ staying between 92−94% in both fetal and adult cultures (Fig. 2C). The average percentage of CD56⁺ cells declined to 71 and 64% in fetal and adult at P6, respectively. However, there was a high donor variability in adult cultures. Out of the four cultures that reached 6 passages, two maintained CD56 expression above 90%, another one—at 50% and the last —at 16%. Notably, one donor of those (A5) retained 98% CD56⁺ cells at 76 doublings (data not shown).

### β-galactosidase associated senescence in expanding adult cells

To test the hypothesis that the fusion index might be associated with the prevalence of senescent cells, β-galactosidase associated senescence was measured qualitatively with β-galactosidase colorimetric staining, and quantitatively using flow cytometry in adult myoblasts ($n = 5$) (Supplementary Data 1). Microscopic analysis revealed the presence of larger cells and, in some donors, an abundance of blue staining in P6 (Fig. 3A). Quantitatively, even though a weak correlation between doubling number and senescence is present (Fig. 3C), the increase of senescence was only observed in some donors (Fig. 3B). The average percent of senescent cells was 34.6 ± 6.9% in P3 and 44.6 ± 20% in P6 with no statistically significant differences between the passages (Fig. 3B). No correlation between fusion index and the percent of senescent cells was found (Fig. 3D).

### Transcriptome differences of low versus highly fusing cells

Bulk RNA sequencing was performed to compare the transcriptomes of proliferating adult LFI myoblasts to HFI myoblasts. RNA samples were collected from myoblast cultures ($n = 5$) at passages 1 and 3, resulting in a total of 10 samples. These 10 samples were ranked from the highest to the lowest fusion indexes, and the top 5 were included in the HFI group, while the last 5 in the LFI group (Fig. 4A). The average fusion index of HFI myoblasts was 32 ± 10%, and of LFI myoblasts 5 ± 6%. This grouping was chosen for better analysis of differentially expressed genes (DEGs) based on the presence of the desired trait (fusion index).

On average, there were $18.7 ± 3.6 × 10^6$ assigned reads per sample. Differential gene expression analysis showed 153 significantly DEGs ($q < 0.05$ and absolute fold change>4) between the two groups, 135 of them were downregulated in LFI cells (215 with fold change < −2) and 18 genes were upregulated (121 with fold change>2) (Fig. 4A, C). Spearman's correlation analysis between z-scores of DEGs and fusion index revealed 8

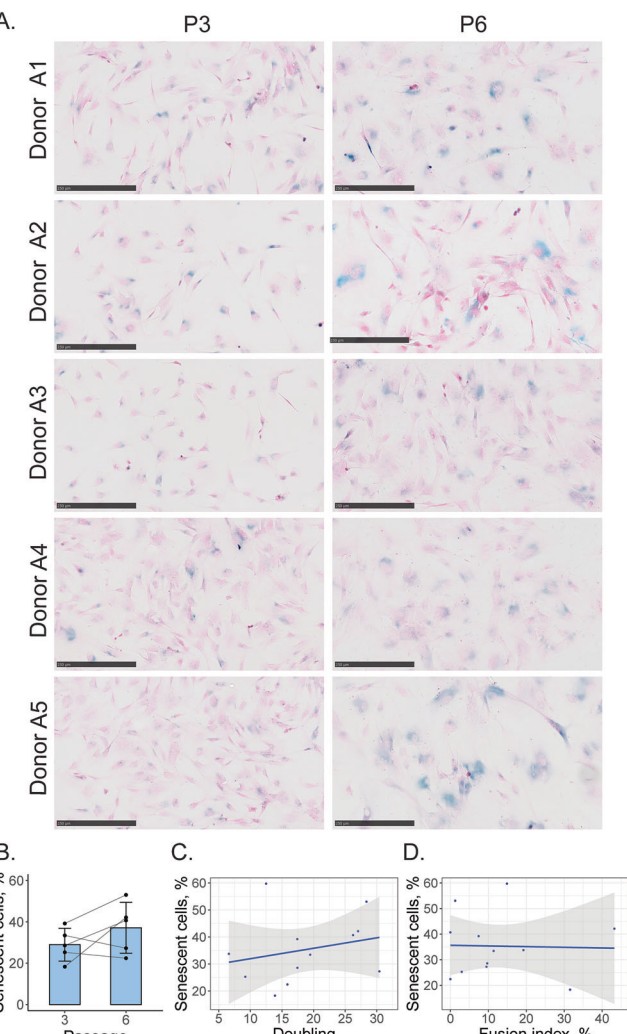

**Fig. 3 | β-galactosidase associated senescence in expanding adult bovine myoblasts. A** Histochemical staining for β-galactosidase (in blue) and cell nuclei (in pink) of proliferating cells. Scale bar = 250 μm. **B** Flow cytometric analysis of β-galactosidase expressing (senescent) cells. Bars are plotted as mean ± SD **C** Weak positive correlation between the percent of senescent cells and their doubling number. **D** Lack of correlation between the percent of senescent cells prior to differentiation and fusion index measured after differentiation.

highly correlated genes with |R| > 0.9 (Fig. 4B) and 30 genes with |R| > 0.8 ($q < 0.05$). Some genes related to G-protein signaling were found to be positively correlated with fusion index (*RGS16, RASL10A*), while others were negatively correlated (*RAB3B, GPR65*). Other genes positively correlated with fusion index were associated with ion transport (*KCNAB1, KCNE4, RYR, NNAT*), modulation of cell adhesion (*VCAM1, CDH17, SDK2, BCAM*) and cytoskeleton (*MAP4, NEB, MYPN*). Notably, they also included oxytocin and myomaker, a gene involved in myoblast membrane fusion. Negatively correlated genes, on the other hand, included several transcription factors (*RARRES1, TFAP2A, HOXD13, PAX2, JAG1*).

To provide an overview of the main affected processes, we first performed g:Profiler analysis to check which gene ontology (GO) terms are overrepresented among the list of DEGs. It found 106 overrepresented biological process-related GO terms ($q < 0.05$) for the downregulated genes (Supplementary Table 4). Among them, g:Profiler showed significant clustering associated with myogenic commitment (e.g., muscle cell development, myofibril assembly, myoblast differentiation), ion transport (e.g., calcium and monoatomic ion transport, regulation of membrane potential), organelle assembly and organization and actin cytoskeleton organization (Fig. 4D). No significant clustering was found among the upregulated genes.

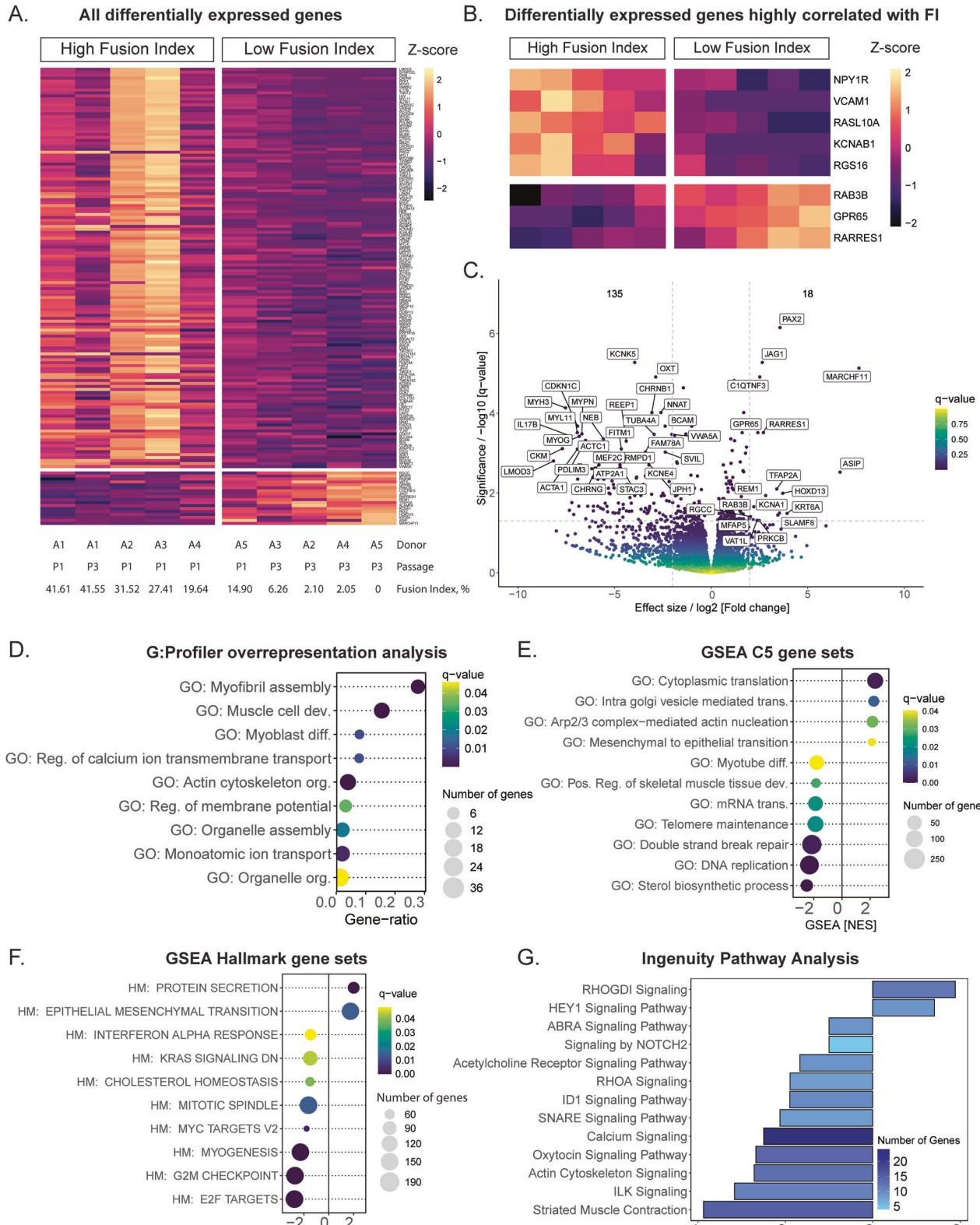

**Fig. 4 | Transcriptomic analysis of proliferating adult bovine myoblasts.**
**A** Heatmap of all genes significantly differentially expressed between the LFI and HFI myoblasts with a description of the samples included in the groups. **B** Heatmap of differentially expressed genes that have the highest correlation with the fusion index (FI) (Spearman's |R| > 0.9, q-value < 0.05). **C** Volcano plot of differential gene expression between LFI and HFI myoblasts. Genes with positive log2 [fold change] are upregulated in LFI myoblasts. The first 30 down- and upregulated genes with the lowest q-value were annotated. **D–G** The significant differentially expressed genes were analyzed using different gene enrichment analysis tools. **D** Selected GO terms of biological processes from G:Profiler overrepresentation analysis. **E** Selected GO terms of biological processes from the GSEA analysis of C5 gene sets. **F** All significantly enriched GSEA Hallmark gene sets. **G** Selection of significant enriched molecular pathways from Ingenuity Pathway Analysis. The full list of the enriched terms is presented in the Supplementary Information. NES = normalized enrichment score.

Then, to further rank the main affected processes, we performed the gene set enrichment analysis (GSEA), which calculated the normalized enrichment scores (NES) taking into account the expression level of each gene. GSEA analysis through the GO term C5 database resulted in total of 4 significant gene sets enriched in LFI myoblasts (NES > 0) and 114 gene sets enriched in HFC myoblasts (NES < 0) (Supplementary Table 5). According to the Hallmark database, there were 2 significant gene sets enriched in LFI myoblasts and 8 in HFI myoblasts (Fig. 4F). The terms enriched in the HFI group included processes related to myogenesis, mRNA transport, DNA replication and double strand break repair, telomere maintenance and sterol biosynthesis (Fig. 4E, F). Among terms enriched in the LFI myoblasts, both sets of C5 and Hallmark genes showed increased protein synthesis (cytoplasmic translation and protein secretion) and mesenchymal to epithelial transition (Fig. 4E, F).

IPA analysis provided a perspective from software with curated databases of molecular pathways. Most canonical pathways are downregulated in LFI myoblasts, consistent with previous observations (Supplementary Table 6). These included pathways related to myogenesis, calcium signaling and cytoskeleton (actin and ABRA signaling), discovered by previous analyzes, but added NOTCH2, oxytocin, ID1, and SNARE. Among upregulated pathways, IPA detected the RHOGDI pathway, driven by the higher expression of *LIMK2* in the LFI cells, and the HEY1 signaling pathway.

The full lists of significantly overrepresented GO terms, enriched gene sets, and signaling pathways can be found in the Supplementary Information (Supplementary Tables 4–6).

## Proteome differences of low versus highly fusing cells

Proteome analysis was performed on both proliferating and differentiated adult myoblasts. Differential analysis of protein expression between proliferating myoblasts with LFI and HFI resulted in the identification of 53,281 peptides (23,048 unique) with a ≥ 95% iProphet probability. Statistical testing and subsequent visual inspection of the volcano plot (displaying log2 fold change and associated *q*-value of each individual peptide) showed 50 downregulated and 3 upregulated peptides, which clearly clustered away from the bulk of the other quantified peptides (Fig. 5A). Due to this clustering, we forewent the usage of a fixed *q*-value cut-off, also as these peptides belonged to the same 11 proteins (8 proteins with peptides less abundant in LFI and 3 proteins with peptides more abundant in LFI) (Supplementary Table 7). Furthermore, all these peptides exceeded a log2 fold change threshold of ±1 (Fig. 5A). Analysis of the proteins associated with the differentially abundant peptides via STRING-db revealed enriched gene sets associated with muscle contraction and DNA repair (Fig. 5B). The associated keyword analysis showed that the most significant protein sets were related to myosin and actin binding and the least significant sets related to methylation and nucleotide binding (Fig. 5C). In the InterPro analysis, however, the XRCC4-like DNA repair protein subunits were both the most abundant and significant, followed by the myosin subunits (Fig. 5D). Analysis of enriched gene ontology terms, however, showed only terms related to actin and myosin (Fig. 5E). The full list of overrepresented GO terms, STRING clusters, UniProt keywords, and InterPro protein domains and features can be found in the Supplementary Information (Supplementary Table 8).

No significant differences or clustering between the LFI and HFI samples in differentiated myoblasts were found (data not shown).

## Discussion

This study evaluated the proliferation and differentiation limits of bovine myoblasts in the context of CM production. While these limitations are generally acknowledged, not many studies take a critical approach to testing them. In primary cell cultures, both donor characteristics and culture conditions influence the quality of cells[11]. In this study, we compared bovine myoblasts from two sources: fetal and adult. When it became apparent that regardless of the source and culture conditions, myoblasts could not differentiate after 25 doublings, we tested several hypotheses on the subset of adult myoblasts to find a possible cause. We also performed transcriptomic

and proteomic analyses of those cells to further characterize this loss of fusion capacity.

As the properties of myoblasts and other adult progenitors can be affected by donor age[31,34,35], we characterized myoblasts from fetal and adult bovine donors. During development from a fetus to an adult organism, cells go through many doublings, and, as a result, adult cells have shorter telomeres needed to protect cells during further doublings[35,36]. Therefore, we hypothesized that fetal cells would intrinsically have a higher proliferation potential. Unexpectedly however, the mean total number of doublings that cells could reach before senescence was lower for fetal myoblasts than for adult myoblasts. This may be due to the culture conditions that were originally developed for the growth of adult cells. For instance, higher oxygen concentrations can lead to oxidative stress of cultured myoblasts[37]. The partial pressure of oxygen (pO$_2$) in fetal bovine blood is approximately 12 mm Hg[38], which is much lower than in adult, approximately 105 mm Hg[39]. Our experiments were carried out under normoxic conditions, resulting in a medium pO$_2$ of 150 mm Hg[40]. Therefore, fetal cells potentially experienced higher oxidative stress compared to their in vivo environment. However, the differentiation capacity of the fetal myoblasts was greater in the first three passages. Despite that, like in adult myoblasts, their fusion capacity was almost completely lost within the first 25 doublings, which is consistent with previous observations[19,21,22]. Nevertheless, these results are promising, because the use of adult cells would be preferred over the use of fetal cells, as it would not require animal slaughter.

We tested two of the commonly proposed reasons for the reduction in differentiation capacity in myoblasts, namely the overgrowth of fibroadipogenic progenitors[21,41] and the onset of replicative senescence[41,42]. Cells were not sorted after isolation, but characterized for the percentage of CD56$^+$ myogenic cells. We observed that the percentage of myogenic cells was initially 85−99% but started to decrease in some cultures at P6 to approximately 64%. However, the fusion index had already dropped from 33 to 13% between P1 and P3, when the percentage of CD56$^+$ cells remained around 90%. This demonstrates the prevalence of myoblasts in early passages and eliminates the hypothesis that the presence of contaminating cell populations is the main reason for reduction in fusion capacity. Moreover, the adult donor with the highest sustained fusion index of ~40%, had the lowest myoblast population of 77%, suggesting that other cell types present in the culture could play an important supporting role in myogenic differentiation[43]. Co-cultures with fibroadipogenic progenitors, vascular cells and smooth muscle cells are sometimes intentionally created for the purpose of achieving better myogenesis and tissue maturation[43]. In addition, co-culture of bovine satellite cells and smooth muscle cells was able to boost myogenin expression[44]. Therefore, the overgrowth of other cell types is not the main driver of reduced myogenesis.

The results on β-galactosidase-associated senescence were inconclusive. We observed an overall high presence of β-galactosidase positive cells (20−50%) and only a slight increase during expansion. However, there was no correlation between β-galactosidase expression in proliferating myoblasts and their fusion capacity. The overall fast doubling rate of the tested cells also contradicts the measurements of senescence. This suggests that this method might not accurately represent the senescence of bovine myoblasts. Especially because β-galactosidase activity is dependent on cell type and species, as pointed out in the original article that reported the use of this marker to measure cellular senescence[45]. Other assays, such as the combination of Ki67 and γH2A.X testing[46], should be considered in the future.

To better understand the molecular mechanisms affected in the cells with reduced fusion capacity, we compared the transcriptomes and proteomes of proliferating LFI and HFI myoblasts from the first passages, when a significant reduction in fusion was observed. Unsurprisingly, many downregulated genes in LFI cells were directly associated with myogenesis, including the genes necessary for myoblast differentiation (*MYOG, MEF2C*), myoblast fusion[47] (*VCAM, CDON, MYMK, ANK3, JSRP1, SYNPO2L*) and sarcomere formation (desmin, myosin heavy and light chain isoforms, actins, nebulin, *KLHL41*). Similarly, there was a higher abundance

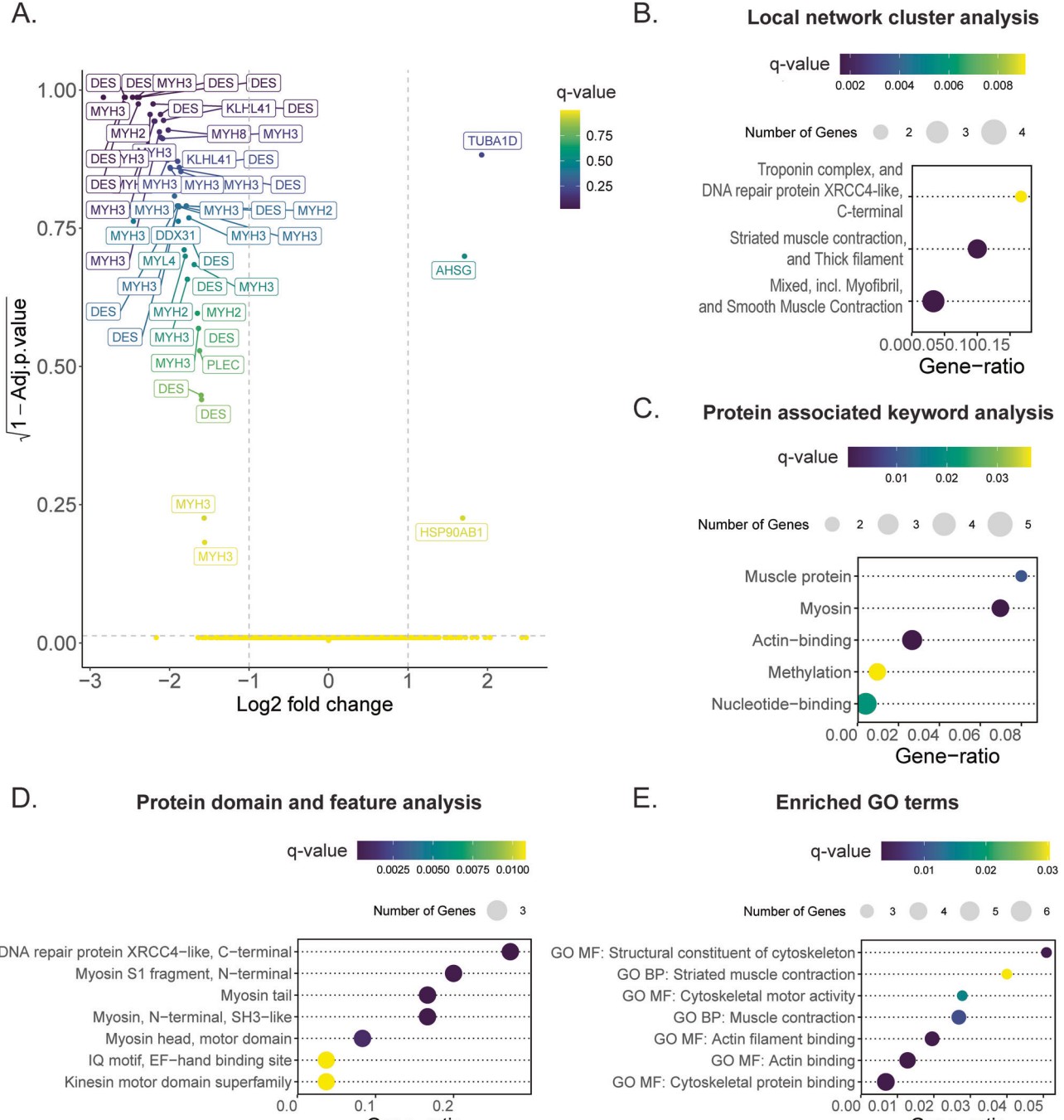

**Fig. 5 | Differential proteomic analysis of adult LFI and HFI myoblasts.**
**A** Volcano plot of differentially abundant peptides. The negative log2 fold change indicates lower abundant peptides in LFI. **B** Local network cluster analysis of differentially abundant peptides using STRING-db. **C** Analysis of protein-associated keywords of differentially abundant peptides using UniProt. **D** Protein domain and feature analysis of differentially abundant peptides using InterPro. **E** Enriched GO terms in differentially abundant peptides.

of proteins associated with sarcomere formation in HFI myoblasts, namely DES, MYH2, MYH3, MYH8, MYL4, and KLHL41. Myogenin was the only downregulated myogenic regulatory factor in LFI myoblasts, but its expression is essential during the later stages of myogenic commitment. Myogenin knock-down bovine myoblasts share many transcriptomic characteristics with our LFI myoblasts, namely gene ontology terms related to sarcomere, myofibril, cell morphogenesis, and cytoskeleton[48]. Myogenin inductor, transcription factor *MEF2C*, was also significantly downregulated, which itself can be repressed by *HEY1* activity, which was increased according to the IPA results. Repression of myogenesis through the HEY1 pathway has been previously reported in C2C12 cells[49]. Therefore, *HEY1* is a

potential molecular target for improving myogenic commitment of myoblasts.

In addition to that, different analyses of both transcriptomic and proteomic data revealed downregulation of DNA replication and repair mechanisms. We did not measure DNA damage in our experiments, but studies in human cells suggest that many DNA breaks can naturally occur during the S phase of cell division in the case of a shortage of proteins required for DNA replication[50,51]. Sometimes these breaks are attributed to the accumulation of reactive oxygen species, but when antioxidant treatment was applied in an attempt to retain the fusion capacity of porcine myoblasts, it led to the opposite outcome[15]. It is possible that the continuous

replication imposed on myoblasts by the growth factors and the stiff tissue culture plastic substrate leads to the accumulation of DNA damage with no time to repair it. In our experiments, almost 90% of DEGs were down-regulated. This can be a result of an underlying process, such as oxidative or replicative stress, leading to untargeted DNA damage and subsequent downregulation of many genes. This suggests that myoblast differentiation capacity is lost in part due to the accumulation of DNA breaks in many important genes caused by replication stress. In fact, periods of quiescence are necessary to maintain the 'stemness' of satellite cells[52]. Furthermore, fast-paced culture inevitably leads to selection of fast-dividing cells, which may be biologically distinct from slow-dividing cells. One study evaluated individual colonies of bovine satellite cells and found that slow-dividing colonies (doubling time >3 days), often had asymmetric division with Pax7$^+$/Myf5$^+$ and Pax7$^-$/Myf5$^+$ daughter cells[34]. However, that study could not show that these cells could form myotubes. Perhaps, traditional culturing methods with continuous cell growth in a medium rich in mitogenic growth factors and passaging should be revised. A new culture system should allow natural asymmetric replication of satellite cells with the period of quiescence[53].

It is unclear whether the observed downregulation of other pathways, such as calcium, oxytocin, and acetylcholine signaling pathways, are the direct consequence of replicative DNA damage, but regardless of the cause, this can have a negative impact on myogenesis. For example, a significantly downregulated calcium signaling pathway, with detected downregulation of voltage-gated channels, could result in low intracellular Ca$^{2+}$ levels. That, in turn, can inhibit myoblast differentiation, as was previously shown with C2C12 cells[54]. However, research that tested increasing intracellular Ca$^{2+}$ uptake with ionophores led to mixed results[55], perhaps due to potential ionophore toxicity[56]. Therefore, manipulation of intracellular Ca$^{2+}$ should be taken with caution. Another pathway, namely the oxytocin signaling pathway, can also be targeted to improve myogenesis. The reduction of oxytocin with age is associated with reduced regeneration of the skeletal muscle, and injection of old mice with oxytocin has been shown to improve myoblast fusion[57]. Aside from downregulated pathways, we also observed upregulation of the RHOGDI pathway, mainly driven by the upregulation of the downstream effector *LIMK2*. LIMK2 is a kinase responsible for inactivation of cofilin, which depolymerizes actin[58]. Actin remodeling is a necessary step during myoblast fusion[26]; therefore, increased activity of LIMK2 can interfere with myogenesis.

Given a high inter-donor variability in gene expression, which impacts DEG analysis, we have also evaluated correlations between individual gene expression and fusion index. We identified 8 genes highly ($|R| \geq 0.9$) and significantly ($q$-value $\leq 0.05$) correlated with the fusion index. A notable example of an inverse correlation is the *RARRES1* gene, which can be upregulated in bovine myoblasts during transdifferentiation to adipocyte-like cells[59]. Therefore, reduced fusion capacity can also be a result of a shift in lineage commitment. *RARRES1* is also downregulated in bull calve myoblasts, which have a better differentiation capacity than dairy cow myoblasts[33]. Notably, when looking at the donor A1, which retained the FI between P1 and P3, not all genes that were correlated with high FI in the whole dataset maintained expression in the P3 of the A1 donor (Supplementary Table 9). For example, downregulation of *RYR1* (calcium transporter in endoplasmic reticulum) and *MYMK* (necessary for myoblast fusion), did not affect the FI of donor A1. While oxytocin, *VCAM1*, potassium voltage-gated channels (*KCNAB1*, *KCNE4*), retained or increased their expression.

Not only does this research help address one of the biggest biological limitations of CM production, it can also provide valuable information to address a common problem in primary cell biology. Almost any practical application of cells, such as stem cell therapies, tissue engineering for regenerative medicine or model tissues, and especially cell-based foods, requires expanding cells to sufficient amounts without compromising on their quality. The significant limitations of primary cells prompted many researchers to shift their focus to pluripotent stem cells[60]. But given that pluripotent stem cell differentiation is often more complex and inferior to primary cells[61], it might still be interesting to continue research on how to overcome the limitations of primary cells. Primary cells lack genetic modification or manipulation and we demonstrated their genomic stability after extensive expansion, making them an attractive candidate for the European cell-based food market[62]. We believe that by providing characterization and omics analysis of bovine myoblasts with reduced fusion capacity, we are able to facilitate research in this direction.

In summary, the present research demonstrates characteristics and temporal changes in in vitro cultured fetal and adult bovine myoblasts with the goal of using them for cultured meat production. Importantly, while the myoblasts of both fetal and adult origin were able to achieve substantial expansion, their capacity to form myotubes at that point was lost. Adult myoblasts in particular demonstrated outstanding expansion capabilities, but special attention should be paid to donor screening due to high inter-donor variability. Further omics analysis of expanding cells revealed multiple affected genes and proteins that are potential targets for improving myoblast differentiation. Follow-up studies can explore inhibition of HEY1 or LIMK2, increasing intracellular Ca$^{2+}$, oxytocin treatment, and mitigating replication-induced DNA damage to determine whether one or several of these solutions can preserve or restore bovine myoblast fusion.

## Materials and methods
### Myoblast isolation and culture
Primary myoblasts were isolated from the biceps femoris of fetal and adult Belgian Blue cattle from a local abattoir within four hours postmortem (Supplementary Table 1). The muscle tissue was transported on ice in high glucose Dulbecco's modified Eagle's medium (DMEM-HG) (Biowest, L0103) containing 2% antibiotic-antimycotic solution (Gibco, 15240062). Upon transportation, the tissue was briefly washed in 70% ethanol and PBS (without Ca$^{2+}$ and Mg$^{2+}$) for disinfection. Then, a 0.5 g biopsy was dissected, avoiding connective tissue and visible blood vessels, weighed, minced, and digested in a GentleMACS Tissue Dissociator (Miltenyi Biotec) using a Skeletal Muscle Dissociation Kit (Miltenyi Biotec, 130-098-305). Erythrocytes were removed with ACK Lysis Buffer for 30 s and neutralized 1 in 10 with 10% FBS in DMEM-HG. Subsequently, the cells were pelleted and resuspended in bovine myoblast proliferation medium+ (PM + ) consisting of DMEM-HG, 20% FBS, 50 µg/ml gentamicin (Gibco, 15750037), 5 ng/ml bFGF (Peprotech, 100-18B) and 10 µM p38 MAPKi (Sigma-Aldrich, SML0543) (Supplementary Table 2). The resulting cell suspension was pre-plated for 1 h in an uncoated T175 flask (Greiner, 660175), after which the supernatant was collected and cultured further on 0.1% gelatin coated T175 flasks in PM + . Where specified, different proliferation media formulations were tested, such as PM, UM, E8, and E8 + BSA (Supplementary Table 2). Freshly isolated myoblasts were labeled as passage 0 (P0). The cells were passaged upon reaching 60−70% confluency with 0.25% Trypsin-EDTA solution (Gibco, 25200056) diluted 1 in 2 with PBS. Following passages were seeded at seeding densities between $2.5 \times 10^3$ and $5 \times 10^3$ cells/cm$^2$.

### Flow cytometric analysis of CD56 marker expression
The proportion of CD56$^+$ cells, representing myoblasts, in isolated cells was characterized using CD56 immunolabeling and flow cytometric analysis. After passaging, $7.5 \times 10^5$ cells were taken for CD56 marker analysis. After washing the cell suspension in 10% FBS in PBS, $2.5 \times 10^5$ of the cells were incubated in APC conjugated anti-human CD56 antibody (Invitrogen, MA1-19462) diluted 1:20 in 10% FBS in PBS, for 25 min on ice in the dark. Another $2.5 \times 10^5$ cells were incubated in the absence of the antibody and used as the Fluorescence Minus One (FMO) control. Lastly, $2.5 \times 10^5$ cells were killed by incubation for 10 min at 65 °C to be used as a control for gating out dead cells. After incubation steps, the cell suspensions were washed 2x in 3% FBS in PBS and resuspended in 100 nM calcein (Anaspec, AS-89201) solution. After 20 min, the CD56$^+$ cells were determined using a CytoFlex S flow cytometer (Analis NV) or FACSVerse™ (BD Biosciences) flow cytometer. At least 10,000 debris- and aggregate-free cells were measured. The gating strategy is presented in Supplementary Fig. 1A.

## Fusion assay

For the fusion assay, $5 \times 10^4$ cells were seeded in gelatin coated 24-well plates and grown in PM for 2-3 days to reach 100% confluency. The medium was then changed to fusion medium SkFM[29], consisting of DMEM-HG, 10 ng/ml hEGF (Peprotech, AF-100-15), 10 µg/ml insulin (Sigma-Aldrich, I9278), 50 µg/ml BSA (Sigma-Aldrich, A2153) and 50 µg/ml gentamicin. Where specified, other fusion media were used, such as FM, consisting of DMEM-HG, 2% horse serum, 50 µg/ml gentamicin; NL15[22], consisting of 1:1 mixture of Neurobasal medium (Invitrogen, 21103049) and L15 medium (Invitrogen, 11415064), 100 ng/ml hEGF, 100 ng/ml IGF (Peprotech, 100-11); or SFDM[24], consisting of DMEM-F12, 10 ng/ml hEGF, 0.5 mg/ml rhAlbumin (Sigma, A9731), 2% ITSE (Gibco, 51500056), 40 µM L-ascorbic acid 2-phosphate (Sigma, A8960), 1 µM LPA (Sigma, L7260), 0.5% MEM AA solution (Gibco, 11130036), 6.5 mM $NaHCO_3$ (Gibco, 11130036), 50 µg/ml gentamicin, 1% soy hydrolysates (Merck, 58903 C)(Supplementary Table 3). After 4 days, or earlier in case the first signs of myotube detachment were observed, the culture was fixed. For the ERK inhibitor experiments, two differentiation protocols were applied: first differentiation protocol was followed as described in publication by Eigler et al. (2021)[30], and second following the protocol described above. Briefly, for the first protocol, $50 \times 10^3$ cells were seeded in a well of 24 well-plate and cultured for 24 h in proliferation medium (PM) consisting of DMEM-HG, 20% FBS, 50 µg/ml gentamicin, then, another 48 h in PM supplemented with 1 µM ERK inhibitor (Santa Cruz, SCH772984), after which the cells were fixed.

Fixation of fusion assay was performed first for 10 min in 4% formaldehyde at room temperature, and then for 10 min in ice-cold methanol at −20 °C, with 3 washes in PBS between the steps. The fixed cultures are then incubated for 1 h in a blocking buffer at room temperature consisting of 1% BSA, 1% Triton-X100 (Sigma-Aldrich, 93443) in PBS, followed by overnight incubation at 4 °C in anti-tropomyosin antibody (Sigma-Aldrich, T9283) diluted 1:100 in PBS. Next, the cultures were washed 3x in PBS and incubated for 30 min at room temperature with Alexa Fluor™ 488 secondary antibody (Invitrogen, A-11029) diluted 1:200 in PBS. After the last 3 washing steps, Hoechst 33342 (Invitrogen, H3570) diluted 1:2000 in PBS was added and the samples were imaged using a Zeiss AxioVert microscope and 10x objective. Six images were taken at the random spots from two wells for each sample as technical replicates.

To analyze the images, we used in-house-developed image analysis software MyoFInDer[63], that determines the proportion of nuclei localized within myotubes to the total number of nuclei. The fusion index is then calculated by dividing this proportion with the CD56+ population analyzed in parallel. The fusion index, therefore, represents the percentage of myoblasts that are able to undergo fusion.

## Cell senescence analysis

Adult myoblasts were subjected to two assays measuring β-galactosidase associated senescence: histochemical staining (qualitative) and flow cytometric assay (quantitative).

The histochemical assay was performed using a Senescence β-Galactosidase Staining Kit (Cell Signalling Technology, 9860S) following the manufacturer's protocol. Briefly, the myoblasts were cultured on gelatin-coated coverslips until reaching 60–80% confluency, fixed, and stained overnight at 37 °C to develop a blue stain in cells which accumulated β-galactosidase. The cells were washed in distilled water and counterstained in Nuclear Fast Red (Carl Roth, N069.1) solution for 5 min. Then, the coverslips were rinsed in distilled water and dehydrated by dipping in a series of ethanol solutions (50, 70, 90, 2x 100%). After the last step of rinsing the coverslips in xylene substitute, the coverslips were air-dried and mounted on glass slides using Sub-X mounting medium (Leica, 3801740). The images were captured using a Nanozoomer microscope (Hamamatsu) and a 20x magnification objective.

Quantitative analysis of senescence was carried out with the CellEvent™ Senescence Green Flow Cytometry Assay Kit (Invitrogen, C10850) following the manufacturer's protocol. In short, $5 \times 10^5$ cells were washed and then fixed in 2% formaldehyde for 10 min at room temperature. After 2 washing steps in 1% BSA, the cells were resuspended in CellEvent™ Senescence Green Probe diluted 1:3000 in CellEvent™ Senescence Buffer and incubated at 37 °C for 1 h in the dark. The negative control consisted of cells incubated in the buffer without the Green Probe. After incubation, the samples were washed in 1% BSA and measured using a FACSVerse™ flow cytometer (BD Biosciences). Ten thousand debris- and aggregate-free events were further gated to isolate singlets. The singlets were then gated in the green fluorescent channel using the negative control to identify the percent of senescent cells. The gating strategy is demonstrated in Supplementary Fig. 1B.

## Genome sequencing

To identify major chromosomal duplications and deletions in expanded populations of bovine myoblasts, shallow genome sequencing was performed. Double stranded DNA was extracted from myoblast cell pellets using DNeasy Blood & Tissue Kit (Qiagen, 69504). The DNA samples were then sequenced using NovaSeq S4 in paired end modus. A minimum of 15 $x10^6$ reads per sample were acquired. The reads were then demultiplexed and aligned to the *Bos Taurus* reference genome (bosTau9, GenBank GCA_002263795.4) using Scalable Nucleotide Alignment Program[64]. The genome was then divided into 250 kb bins to compare the number of copies of each bin. Corrected log2 mean read depth ratios were plotted across the genome to find any chromosomal aberrations.

## Transcriptome analysis

Pellets of $1 \times 10^6$ cells were washed in PBS and frozen without supernatant at −80 °C for storage. Upon thawing, the cell pellets were processed using the Pure Link™ RNA Mini Kit (Thermo Fisher, 12183018 A) to extract the RNA and treated with PureLink™ DNase (Thermo Fisher, 12185010) to remove DNA from the samples. The RNA library was prepared with the QuantSeq 3′ mRNA-Seq kit (Lexogen) and the samples were sequenced using high-output HiSeq 4000 (Illumina) with a read length of 51 bp. Quality control of raw reads was performed with FastQC v0.11.7, and the adapters were filtered using Trimmomatic v0.39. Splice-aware alignment was carried out using Hisat2[65] against the ARS-UCD1.2 reference genome using the default parameters. Quantification of reads per gene was conducted using featureCounts from the Subread package[66]. Count-based differential expression analysis was performed using the R-based Bioconductor package DESeq2[67]. The library preparation and sequencing were performed by Genomics Core Leuven.

Significant DEGs were filtered using a *q*-value cutoff of 0.05 (Benjamini-Hochberg procedure) and an absolute log2 fold change cutoff of 2. Gene expression of LFI cells was compared to HFI cells, meaning that a positive fold change represents an upregulation in LFI cells and negative—downregulation in LFI cells. To represent the expression of these significant genes across samples and conditions, Z-scores were calculated for each gene according to the formula: $Z = (x - \mu)/\sigma$, where $x$ is a normalized count of a gene, $\mu$ is a population mean of the normalized counts and $\sigma$ is a standard deviation of the normalized counts in the population.

An overrepresentation analysis of the filtered DEGs was done using the R based g:Profiler package[68], including only GO terms from Biological Processes database. GO terms were first reduced using a *q*-value cutoff of 0.05 (Benjamini-Hochberg procedure) and a minimum of 3 genes involved. Then the GO terms were further reduced using the rrvgo package to better summarize the GO sets based on their functionality[69]. R scripts for g:Profiler analysis and the necessary datasets can be accessed via Zenodo[70].

Additionally, Gene Set Enrichment Analysis (GSEA, v4.3.3)[71] was performed on the normalized count data from the DEGs using default settings. The datasets used for this analysis were sourced from the Molecular Signatures Database, specifically C5 gene set that consists of standardized GO terms (c5.go.bp.v2023.2.HS.symbols.gmt) and the Hallmark gene set (h.all.v2023.2.Hs.symbols.gmt). Gene sets were filtered based on a *q*-value cutoff of 0.05.

QIAGEN's Ingenuity Pathway Analysis (IPA) software (version 111725566) was used to perform the (canonical) pathway analysis on the

non-filtered DEGs. DEG cutoffs for IPA were set at an absolute log2 fold change of 1 and q-value of 0.05. Pathways were filtered based on their -log (q-value) (q < 0.05) adjusted with Benjamini-Hochberg procedure.

## Proteome analysis

Proliferating (non-differentiated) myoblasts were cultured to confluency prior to protein sample collection, while differentiated myoblasts were additionally cultured in the fusion medium for 4 days prior to sample collection. The cells were washed with PBS, treated with ice-cold protein precipitant consisting of 20% trichloroacetic acid (Carl Roth, 3744.3), 80% acetone, 0.2% dithiothreitol (Carl Roth, 6908.1), and scraped from the tissue culture plastic. The cell-bearing precipitant was then incubated overnight at −20 °C, after which the content of the tubes was pelleted at −9 °C, 17,000 g for 25 min. The supernatant was discarded and the pellets were washed for 2 h at −20 °C in ice-cold 80% acetone. After another centrifugation step and complete removal of the supernatant, the pellets were air-dried for 10 min and resuspended in urea lysis buffer consisting of 8 M urea and 50 mM ammonium bicarbonate. The samples were then sonicated using a 60 W sonic dismembrator following 6 cycles of alternating pulsing (30 s of 0.3 s on, 0.7 s off) and resting (1 min), with samples kept in ice water during the procedure. The samples were then centrifuged at 16,000 g at 25 °C for 10 min and the protein concentration in the supernatant was determined using a Roti®Quant universal assay (Carl Roth, 0120.2).

Thirty µg protein extract of each sample was reduced and alkylated using 10 mM tris(2-carboxyethyl)phosphine hydrochloride (Sigma-Aldrich, C4706-2G), 25 mM chloroacetamide (Sigma-Aldrich, C0267-100G) for 45 minutes at room temperature in the dark. Afterwards, the samples were diluted 1:4 in 50 mM ammonium bicarbonate (Sigma-Aldrich, 09830-500 G) and 800 ng Trypsin (Pierce, 90057), resuspended in 50 mM ammonium bicarbonate, was added. Proteins were digested overnight at room temperature. The next day, trifluoroacetic acid (Merck, 1.08262.0025) was added to a total concentration of 1% (v/v) to stop the digestion and peptides were cleaned up using a home packed C18 column (Empore SPE Disks, 3 M, 66883-U). Afterwards, peptides were snap-frozen and freeze-dried (Alpha2-4 LSC, Martin Christ Gefriertrocknungsanlagen GmbH). Peptides were resuspended in 50 mM HEPES (Sigma-Aldrich, 54457-10G-F) and 24% acetonitrile (Merck, 1.00029.1000)[72]. TMT10plex (Lot No.: UE282321, Thermo, 90110) was resuspended in 52 µL 100% anhydrous acetonitrile (Sigma-Aldrich, 271004-100 ML). One set of TMT10plex was used to label the 10 non-differentiated and 10 differentiated samples, respectively. For this, 2.32 µL of the respective label was added to the samples and incubated for 1 h at room temperature ( = 30% final acetonitrile concentration during the labelling reaction). The labelling reaction was stopped by adding hydroxylamine (Sigma-Aldrich, 467804-10 ML) to a final concentration of 0.5% (w/v). After incubation for 15 min at room temperature, the respective set of the proliferating and differentiated myoblast samples was combined, snap-frozen, and freeze-dried. Samples were afterwards resuspended in 2% (v/v) acetonitrile (Merck, 1.00029.1000), 0.1% (v/v) formic acid (Merck, 5.33002.0050) and analyzed by LC-MS/MS.

For LC-MS/MS analysis, the peptides were re-dissolved in 20 µl solvent LA (0.1% trifluoroacetic acid in water/acetonitrile (ACN) (98:2, v/v)) and 15 µl were injected on an Ultimate 3000 RSLCnano system which was in-line connected to a Q Exactive HF mass spectrometer (Thermo). Peptides were loaded onto a 5 mm trapping column (Thermo scientific, 300 µm internal diameter (I.D.), 5 µm beads) for 2 min in solvent LA and a flow-rate of 20 µl/min. The peptides were separated on a 250 mm Aurora Ultimate, 1.7 µm C18, 75 µm inner diameter column (Ionopticks), for which the temperature was set at 45 °C. Peptides were eluted by a non-linear gradient using a two-solvent system (solvent A: 0.1% FA in water; solvent B: 0.1% FA in water/acetonitrile (2:8, v/v)) from 0.5% solvent B to 33% solvent B in 135 min, then an increase to 55% solvent B over 20 min and, 70% solvent B in 5 min. This was followed by a 5 min wash at 70% solvent B and re-equilibration of the column with solvent A. The mass spectrometer was operated in data-dependent mode and full-scan MS spectra (350–1500 m/z) were acquired in the Orbitrap analyzer with an AGC of $3 \times 10^6$ and a

resolution of 60,000. The 12 most intense ion species above a threshold intensity value of 15,000 were isolated with a width of 1.5 m/z for HCD fragmentation (NCE: 33%), after filling the trap at a target value of 13,000 for a maximum of 80 ms, and a resolution of 45,000 in the Orbitrap analyzer. The mass spectrometry raw files were converted to mzML format using ProteoWizard (Version: 3.0.23301-4fd35e2)[73]. Raw data was searched using Comet (version 2023.01 rev. 2)[74] against the UniProt Bos Taurus reference proteome FASTA (UP000009136_9913, 26,635 protein entries, downloaded 3rd June 2024 from www.uniprot.org). As variable modifications oxidation of methionine was set, fixed modifications included carbamido-methylation of cysteine, and TMT10plex label on lysine and peptide and protein N-termini. As protease Trypsin without proline rule was set, allowing for 2 missed cleavages. Precursor mass tolerance was set to 20 ppm, the fragment bin tolerance was set to 0.02 and the fragment bin offset to 0.0 (as per default values for high resolution MS/MS), additionally the TMT reporter ion mass range (126.0 - 131.3 m/z) was excluded from the mass list during peptide identification. Resulting pep.xml output files were further processed using PeptideProphet[75], iProphet[76] and ProteinProphet[77] as part of the Trans-Proteomic Pipeline (version 6.3.0 Arcus, Build 202305041110-8946)[78]. In the case of PeptideProphet, accurate mass binning using ppm, the usage of decoy hits for characterization of the negative distribution, and the usage of the non-parametric model and the reporting of the computed probabilities for the decoy hits were set. For iProphet and ProteinProphet, the default settings were used. After analysis, peptides were filtered with an iProphet probability cut-off of 95% ( = 5% FDR).

TMT reporter ion intensities were extracted from the mzML raw files and corrected for isotope errors using R (version 4.1.3) via home-made scripts. The data analysis scripts are based on scripts used in Khanam et al.[79]. In case of peptides which were measured at least twice and exhibited the same sequence and modifications, the raw intensity data were averaged. Afterwards, the TMT reporter intensity data was log2-transformed.

## Statistics and reproducibility

All data was analyzed and visualized using RStudio (version 4.3.1). Normality of the data was checked with a Shapiro test. For group comparisons, homogeneity of variance was checked with a Levene's test. Parametric statistical tests were performed on normally distributed data. Data is reported as mean ± standard deviation, unless stated otherwise. Statistical significance cutoffs are set at $p = 0.05$ (*), $p = 0.01$ (**), $p = 0.001$ (***). In transcriptomic and proteomic analyses, the reported p-values underwent adjustment for multiple testing to control the false discovery rate (FDR) and are reported as q-values. Where time variable is an independent variable (passage, doubling), it is implied that the same samples are measured repeatedly to show the changes in dependent variables. The exceptions are the samples collected for the transcriptome and proteome analysis, which are described in the corresponding results section.

For proteomics, the normality of the data distribution was assessed using QQ-plots. Statistical testing was carried out using limma[80] with enabled robust hyperparameter estimation[81]. Peptides were visualized using a volcano plot, and in the non-differentiated samples a group of 53 peptides (from 11 proteins) clustered away from the bulk of all quantified peptides and were thus regarded as peptides of interest. These peptides of interest were further analysed via string-db (https://string-db.org/)[82] (local network cluster analysis (STRING), protein domains and features (InterPro), GO terms and annotated keywords according to UniProt. The following R libraries were used: dplyr[83], extrafont[84], ggplot2[85], ggpointdensity[86], ggrepel[87], limma[80], matrixStats[88], MSnbase[89], plyr[90], reshape2[91], scales[92], stringr[93], viridisLite[94], wesanderson[95], XML[96], xml2[97]. All mass spectrometry raw data and search results were uploaded to jPOSTrepo[98] and can be accessed via ProteomeXchange accession number PXD052959. Data analysis scripts can be accessed at Zenodo[70].

Our study presents results from 5 adult and 5 fetal bovine donors (n), as a compromise between higher statistical power and limited resources. Therefore, n represents biological replicates/donors throughout the study. Additional experiments were presented in the supplementary material with

various sample sizes ($n = 2$–5). No power tests were performed to determine the right sample size, but rather a few donors were tested to help selection of the culture conditions for further experiments presented in the study. No data were excluded. Some of the variables (such as cell doubling numbers, fusion index, CD56 marker analysis) were analyzed independently for the same samples in other unrelated experiments, are were showing similar results, and therefore, indicating reproducibility. Additionally, the instruments used for data acquisition are regularly maintained and pass quality controls. Bovine donors for muscle tissue collection were randomly selected and were only based on availability at the local abattoir. The investigators were not blinded during data collection and analysis to avoid mistakes during the execution of the experiments. Potential bias was minimized by using data processing software were possible (e.g. fusion assay image analysis, flow cytometry).

## Reporting summary
Further information on research design is available in the Nature Portfolio Reporting Summary linked to this article.

## Data availability
Source data for all graphs presented in the manuscript can be found in Supplementary Data 1 and Supplementary Data 2. Processed RNA sequencing data in form of global normalized counts is accessible via Zenodo repository (https://doi.org/10.5281/zenodo.13897183). Raw and processed mass spectrometry data for proteome analysis can be accessed under PXD052959. Any additional raw or processed data are available upon request.

## Code availability
Data analysis scripts for the proteome and transcriptome analysis were made available via Zenodo repository (https://doi.org/10.5281/zenodo.13897183).

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

## Acknowledgements

This work was conducted with financial support of FWO-SBO Customeat (S002821N) and KUL KA/22/042. The authors thank the Genomics Core Leuven for performing bulk RNA-sequencing and shallow genome sequencing, Björn Menten from the Lab of Genome Research at Ghent University for performing shallow genome sequencing data analysis, Valerie Begun, Aaron Oosthuyse, and Lander Rabaut who assisted with the execution of the experiments.

## Author contributions

M.O. and L.T. contributed to conceptualization and experimental design. M.O. and F.W. executed the experiments and collected the data. M.O., A.B., and F.W. contributed to data analysis, interpretation, writing of the original draft and manuscript revisions. C.P. and L.T. contributed to project supervision and revision of the manuscript. L.T. contributed to acquisition of the funding for the project.

## Competing interests

The authors declare no competing interests.
