## [Transparent Peer Review file · Communications Biology]

Addressing hurdles in cultured meat by exploring the mechanisms behind reduced myogenesis after bovine myoblast expansion

Corresponding Author: Professor Lieven Thorrez

Version 0:

Reviewer comments:

Reviewer #1

(Remarks to the Author)

The manuscript titled "Addressing hurdles in cultured meat by understanding the mechanisms behind reduced myogenesis after bovine myoblast expansion" is well-written and contains interesting data to the field of cultivated meat and cell biology research in general. The work is convincing and has some original conclusions. There are a few limitations, questions and unexplored ideas, raised below, that could be commented upon or adjusted to further improve the manuscript. Some questions are also just for the sake of reflection and of genuine interest that I hope the authors can supply answers to.

1) There is no mention on the use of serum-free media. The study uses 20% FBS which is quite high. Also, the media contains a p38 inhibitor, presumably to limit spontaneous differentiation, which should be commented on. Could serum-free media affect the long-term myogenic performance? At least using serum-free media can delay differentiation (REF 1). What about a more optimized differentiation media – maybe using Erk inhibitor (REF 2) to induce it in the P6 samples?

2) What is the argument for doing the omics data on proliferating cells and not cells that are undergoing myogenesis? I understand the reason for the experimental design in which the high fusion and low fusion donors were grouped for the transcriptomics and proteomics data. However, it would be equally (if not more) interesting to look at passage 1 vs passage 6 to see how the individual donors changed over time because they all lose their myogenic capacity at that point (except one donor). And now mentioning this one donor that performs well at P6, why not single this one out in the transcriptomics data (maybe just suppl.) to see if there is anything wildly different in this one? Does it still retain its fusion beyond P6? In the same manner, do you have any individual data for the good performing donor in terms of proliferation (Donor A5)? Also, where there any correlation between donors that performed well in proliferation vs differentiation?

3) The conclusion is lacking a bit of commenting on the evidently very good proliferative capacity (at least some donors) and the difference between fetal and adult cells. Maybe circle back to figure 1A and state something about this. The fact that adult cells are potentially better long-term is actually very nice! This means we can use biopsies throughout the animals life or use tissue from slaughtered adult animals. The conclusion must also reflect the main issues in using primary cells for cultivated meat, which is the donor variation (limited proliferation in some donors) and then the main finding of very little differentiation after P3, which did not have any clear solutions in this study. Also, is differentiation even necessary for cultivated meat? e.g., companies are using fibroblasts without any myogenic capabilities.

4) Reference format is not consistent, e.g. all names and et al. both are being used.

5) Passaging: what is the T-flask format and what are the cell densities being used? It might also be nice to have the accumulated cell numbers for some of the good performers to have as a reference.

6) Is CD56 actually the best surface marker when it comes to FACS? (REF 3)

7) The fusion assay is performed "after 4 days or when myotubes detachment is observed". This statement is a little bit concerning. This means that there are potentially some samples which has lost their myotubes affecting the data. Also, the loose timeframe 4 day + xx days could also skew data in a certain way. The way fusion is calculated is stated as the

proportion of CD56 positive cells within myotubes – this could skew results in samples with lower CD56 positive populations (e.g. the high fusion performer which is stated to have the lowest CD56 population 77%). Why not the normal way with just the myonuclei in relation to total nuclei? The fusion data is based on the average of 6 images – are there there replicate wells as well? Standard deviation from these replicates? How do you define “random” in the image selection?

8) Why is there no mention of the upregulated TUBA1D and AHSG in the low fusion samples in the discussion?

Line 75: two studies to your knowledge. REF 1 also has that comparison.

Line 346: these two formula are weird to present in the main text. Maybe put it into figure 2B.

Line 517-520: the reference is a review – what is the actual piece of research showing this?

Line 520-521: in the referenced study is actually just the smooth muscle cells co-culture that boosts myogenin – the tri-culture compared to this one is non-significant. Also, more myogenin expression ≠ better myogenesis.

Line 559: how can you conclude this based on the 90% DEGs being downregulated?

Line 590: RARRES1 genes has also been shown to be less expressed in bull calf samples vs dairy cow samples (bull calf showed better myotube formation)(REF 1).

Line 602: should it not be genomic stability?

REF 1: <https://doi.org/10.1016/j.foodres.2023.113217>

REF 2: [10.1016/j.devcel.2021.11.022](https://doi.org/10.1016/j.devcel.2021.11.022)

REF 3: <https://doi.org/10.3389/fnut.2023.1212196>

Reviewer #2

(Remarks to the Author)

Overall, the presented work by Olenic et al. includes a couple of interesting and potentially unexpected observations. The first is that there is not a major difference in performance between cells derived from fetal and adult cows. The second is that for at least a subset of myoblast cell cultures, proliferation beyond the hypothetical ‘Hayflick limit’ was observed. The general observation that primary cell cultures show reduced proliferation and differentiation over time is fairly well established at this point.

Overall, whilst these are interesting initial observations, in my opinion they are not yet a complete story that adds significantly enough to the cultivated meat field to warrant publication in a journal such as Communications Biology.

Comments:

The use of serum-containing medium for cell proliferation (and differentiation) is quite out-of-date for the field by this stage, and there are a number of formulations (Stout 2022, Kolkmann 2022, Messmer 2023) that could be tested (some of which the authors cite). It is known that serum contains a lot of highly pro-proliferative factors, and whether the continued proliferation would also be observed in serum-free medium is unclear. This study mirrors the Ding et al 2018 paper quite closely, but that article attempted some mechanistic explanations of satellite cell aging in vitro and included small molecule inhibitor testing that is not present here.

The beta-galactosidase experiments presented in Figure 3 are very unconvincing, it’s not clear to me whether the staining is even specific. Comparing only two passages, when the cells showed well over 50 cumulative population doublings, also seems very minimalist.

The OMICs experiments, whilst nicely performed, are very descriptive, providing little mechanistic insight into why different behaviours emerge, because there is no way to distinguish between cause and effect in terms of cell aging and loss of myogenic potential. In my mind, a more interesting transcriptomic comparison would be between old vs young cells, to give insight into why some cell lines are able to continue proliferating and to potentially inform strategies for medium or cell development to improve future cultures. The title of the article claims that this study helps to “understand the mechanisms behind reduced myogenesis after bovine myoblast expansion”, but I don’t really see how, beyond the immediate observation that better differentiating cultures show higher expression of myogenic genes.

No flow cytometry plots are presented in the main figures or supplementary data, making it impossible to ascertain the quality of these data and independently assess the authors claim that overgrowth of different cell types is not a factor in the continued proliferation.

The doubling time below which cultures were excluded from subsequent experiments is set by rather arbitrary cut offs.

The discussion is extremely lengthy (longer than the results section itself) and hence starts to stray towards speculation that goes quite far beyond the actual results presented in this study.

Overall, the data is generally of a good quality (Figure 3 aside), but in my opinion this represents a promising set of observations to start off a project, rather than a complete work, and hence would need substantial further experiments to be considered for publication.

Version 1:

Reviewer comments:

Reviewer #1

(Remarks to the Author)

Dear Authors,

Well done on answering my questions and addressing the concerns I had mentioned. There are certain things we might disagree on, yet your argumentation is fair within the context of this study.

The manuscript now stands much stronger, especially with the additional supplementary data. In my opinion it is very important to also address the poor performance of all the different serum-free media types that you have used. We need to learn more about why they sometime perform so bad to improve them. The fact that you have tested a lot of different media types also validates your approach with the serum-containing media. It was simply necessary in your case.

There are still limitations of course, yet with the new additions and further acknowledgement of said limitations, I am convinced that the manuscript now delivers interesting and scientifically sound results that does not over-extrapolate. This is also clear now with the changed title.

Conclusion:

The data highlights an already-known issue in muscle stem cell culture, with some new insights and give grounds for a needed discussion on this topic. The novelty might not be incredibly high, however the work is convincing and results like these with fundamental cell biology character are much needed in the cultivated meat community. I do not have any further comments or suggestions for improving the manuscript beyond its current state, which I find satisfactory.

Reviewer #2

(Remarks to the Author)

Overall, the revised version of the manuscript is slightly more nuanced, and the title change is appreciated. However, in this reviewers opinion it does not contain any fundamentally new insights or experiments when compared to the initial version, and therefore remains good quality but incomplete work.

The authors claim that accumulation of DNA damage might underlie differences between their cultures, but no experiments, which would be fairly simple to conduct, are offered to back this up, nor any other hypothesis from the OMICs data.

Response letter – COMMSBIO-24-6780-T

We thank the reviewers for their time to review our manuscript as well as their valuable suggestions. In this revision we would like to provide the requested additional data and provide answers to the raised questions.

Both reviewers raised concerns about the culture medium used in our experiments because it contains serum, while a number of publications report the use of serum-free medium. There are two reasons why serum-free medium was not used in this work. Firstly, despite trying several published serum-free culture medium formulations, we were not able to expand bovine myoblasts in them to quantities sufficient for the selected experiments (results presented below). Secondly, adapting published serum-free media formulations was outside of the scope of our work. But we certainly agree that adaptation of serum-free media in cultured meat research and development is needed and hope that commercialization of existing formulations will improve accessibility and robust reproducibility of the published results.

Reviewers' comments:

Reviewer #1 (Remarks to the Author):

The manuscript titled "Addressing hurdles in cultured meat by understanding the mechanisms behind reduced myogenesis after bovine myoblast expansion" is well-written and contains interesting data to the field of cultivated meat and cell biology research in general. The work is convincing and has some original conclusions. There are a few limitations, questions and unexplored ideas, raised below, that could be commented upon or adjusted to further improve the manuscript. Some questions are also just for the sake of reflection and of genuine interest that I hope the authors can supply answers to.

1) There is no mention on the use of serum-free media. The study uses 20% FBS which is quite high. Also, the media contains a p38 inhibitor, presumably to limit spontaneous differentiation, which should be commented on. Could serum-free media affect the long-term myogenic performance? At least using serum-free media can delay differentiation (REF 1). What about a more optimized differentiation media – maybe using Erk inhibitor (REF 2) to induce it in the P6 samples?

We thank the reviewer for bringing up the use of serum-free medium. We previously tried different formulations of the serum-free media published in the cultured meat field, namely E8 (10.1007/s10616-019-00361-y), SFGM (10.3389/fnut.2023.1212196), Beefy-9 (10.1007/s10616-019-00361-y), and Beefy-R (10.1016/j.biomaterials.2023.122092). Unfortunately, we were not able to expand bovine myoblasts sufficiently in any of the tested formulations. In SFGM, Beefy-9 and Beefy-R our fetal cells failed to reach confluency (Response Letter Figure 1, exp. #1). Adult cells were growing slower in serum-free growth media and failed to re-attach upon passaging (Response Letter Figure 1, exp. #2), therefore, any further analysis was impossible. We did not test the medium provided in REF 1, as this medium was not tested in long-term cultures (also stated in the limitations of the study).

Figure 1

However, to address the reviewers question about p38 inhibitor, and to report on results we obtained with serum-free E8 medium, we decided to include additional results (Supplementary Figure 2 and 3) about:

- Experiments, where p38 MAPKi was added to the proliferation medium (PM vs PM+) to prevent premature differentiation during expansion
- Experiments where 50% of serum was replaced with a serum replacement Ultroser
- Experiments with E8 medium based on the performance of this medium in Kolkmann et al., 2020 and its iterations to improve cell adhesion, proliferation and differentiation

However, we would like to emphasize that the goal of this paper was not to find optimal cell culture conditions or find serum-free alternatives. The context in which PM+ medium was selected for the further experiments has now been highlighted in Supplementary Figure 2 and lines 377-385.

Additionally, we added Supplementary Figure 4, where we report on all previously tested differentiation medium formulations, including SFDM, NL15 and a medium containing 2% horse serum differentiation medium commonly used in literature. These results explain our choice of differentiation medium (SkFM) (Gholobova et al., 2015). Furthermore, we performed more experiments including ERKi supplementation during myoblast differentiation that showed that ERKi supplementation is ineffective in inducing fusion in myoblasts with reduced fusion capacity (Supplementary Figure 4: E-M).

References:

Kolkman AM, Post MJ, Rutjens MAM, van Essen ALM and Moutsatsou P 2020. Serum-free media for the growth of primary bovine myoblasts. *Cytotechnology* 72, 111–120.

Gholobova D, Decroix L, Van Muylder V, Desender L, Gerard M, Carpentier G, Vandeburgh H and Thorrez L 2015. Endothelial network formation within human tissue-engineered skeletal muscle. *Tissue Engineering - Part A* 21, 2548–2558.

2) What is the argument for doing the omics data on proliferating cells and not cells that are undergoing myogenesis? I understand the reason for the experimental design in which the high fusion and low fusion donors were grouped for the transcriptomics and proteomics data. However, it would be equally (if not more) interesting to look at passage 1 vs passage 6 to see how the individual donors changed over time because they all lose their myogenic capacity at that point (except one donor). And now mentioning this one donor that performs well at P6, why not single this one out in the transcriptomics data (maybe just suppl.) to see if there is anything wildly different in this one? Does it still retain its fusion beyond P6? In the same manner, do you have any individual data for the good performing donor in terms of proliferation (Donor A5)? Also, where there any correlation between donors that performed well in proliferation vs differentiation?

Thank you for raising good points! We only did transcriptomic analysis of undifferentiated myoblasts to find the gene expression that is pre-requisite for fusion capacity. It would be no surprise to find reduced expression of sarcomeric proteins of differentiated samples with less myotubes. Therefore, we found it more valuable to look at the molecular processes happening inside of myotube precursor – myoblasts.

Regarding the donor that preserved its fusion capacity till passage 6, indeed, we also previously had an idea to shortlist the genes which were differentially upregulated or downregulated in this sample, labelling them as potentially “necessary” or potentially “redundant” for fusion. However, we previously had some doubts about how representative this case study would be, and whether it would be reproducible. Nevertheless, we now included these results in Supplementary Table 9 and discussed them in lines 680-686.

We did not observe any correlations between fusion capacity (FI) in the beginning of the expansion (passage 1) and proliferation capacity (either total number of doublings or doubling time) as can be seen in the Pearson’s correlation plots presented below (Response Letter Figure 2 and 3).

Figure 2

Figure 3

3) The conclusion is lacking a bit of commenting on the evidently very good proliferative capacity (at least some donors) and the difference between fetal and adult cells. Maybe circle back to figure 1A and state something about this. The fact that adult cells are potentially better long-term is actually very nice! This means we can use biopsies throughout the animals life or use tissue from slaughtered adult animals. The conclusion must also reflect the main issues in using primary cells for cultivated meat, which is the donor variation (limited proliferation in some donors) and then the main finding of very little differentiation after P3, which did not have any clear solutions in this study. Also, is differentiation even necessary for cultivated meat? e.g., companies are using fibroblasts without any myogenic capabilities.

Thank you for these insightful suggestions. We have emphasized the advantage of using adult cells in our updated discussion in lines 588-589. We have also addressed the issue of donor variability in lines 703-705. Regarding the question if it's even necessary to differentiate the myoblasts for cultured meat, we believe that it is out of the scope of this paper to delve into this philosophical discussion. Proliferation and differentiation of myogenic cells still remain the desired way to produce cultured meat to be as biologically close to meat as possible, and was discussed in the introduction in lines 38-46.

4) Reference format is not consistent, e.g. all names and et al. both are being used.

References are formatted according to Nature's citation style, which shortens the author list to first author + et al. in case of more than 5 authors.

5) Passaging: what is the T-flask format and what are the cell densities being used? It might also be nice to have the accumulated cell numbers for some of the good performers to have as a reference.

We now clarified the T-flask format (T175) and seeding density (2500-5000 cells/cm²) in the corresponding M&M section.

Instead of cell numbers, we focused on the number of doublings, linked to cell numbers in Figure 1 panel A and in lines 47-59, because it is easier to follow and more representative of cell proliferation.

6) Is CD56 actually the best surface marker when it comes to FACS? (REF 3)

Our group has been using CD56 as a marker for human myoblasts for about a decade. For human cells, the expression of CD56 is coinciding with desmin expression. For desmin, we used immunocytochemistry, but CD56 as a cell surface markers has been used in parallel and since FACS analysis is more rapid than immunocytochemistry, CD56 has been used as the primary marker. When we started to work with bovine cells (in 2019), desmin immunocytochemistry did not work well, so that's why we focused on CD56. The bovine CD56 expression in the early passages correlates well with the fusion capacity, underscoring its relevance as a marker for bovine myoblasts. Moreover, REF 3 (Messmer et al. 2023) confirms that CD56 (=NCAM1) is the best marker that is present on SC (satellite cells), but not or much lower on FAPs, VECs, LECs, SMMCs, monocytes/macrophages and glial cells (figure 2).

7) The fusion assay is performed "after 4 days or when myotubes detachment is observed". This statement is a little bit concerning. This means that there are potentially some samples which has lost their myotubes affecting the data. Also, the loose timeframe 4 day + xx days could also skew data in a certain way. The way fusion is calculated is stated as the proportion of CD56 positive cells within myotubes – this could skew results in samples with lower CD56 positive populations (e.g. the high fusion performer which is stated to have the lowest CD56 population 77%). Why not the normal way with just the myonuclei in relation to total nuclei? The fusion data is based on the average of 6 images – are there there replicate wells as well? Standard deviation from these replicates? How do you define "random" in the image selection?

In the fusion assay we typically fixate and stain the cells after 4 days in the differentiation medium, or earlier if the first signs of myotube detachment are observed. Myotube detachment is a rare occurrence in myoblasts with fusion index approaching 100% and is very evident under microscopic observation (Response Letter Figure 4). To calculate the fusion index of such cultures, we then avoid taking pictures at the spots where myotube detachment is seen. In this way, we are still able to quantify the fusion index without underestimation due to myotube loss. To clarify that we adjusted the text in line 159.

Figure 4

Regarding the formula of fusion index, it is deliberately calculated as a proportion to CD56+ cells, because we specifically want to report the proportion of myoblasts that fuse. A presence of contaminating cell populations is one of the most obvious reasons for reduced overall fusion, therefore we considered it important to correct the results for it to report on the differentiation potential of the myoblasts, which the manuscript focuses on.

Six images were taken across two replicate wells. These were pooled as technical replicates. We now added this specification to the revised version of the manuscript (lines 176-177). The standard deviation of technical replicates was typically around 5%, much lower than standard deviation between biological replicates.

“Random” means that the images were not selected based on presence or absence of myotubes. In fact, to make this process more blind, we used the DAPI channel to select a spot, rather than the green channel, where the myotubes could be seen.

8) Why is there no mention of the upregulated TUBA1D and AHSG in the low fusion samples in the discussion?

TUBA1D is the alpha-1D chain of tubulin, a major constituent of microtubules. AHSG is also known as fetuin A. Both were indeed found upregulated in low fusing myoblasts. However, at present we do not see any clear mechanistic explanations for these observations. We agree it may be of interest to further study the role of these proteins in bovine myoblast differentiation, however, that was outside the scope of the current study.

Line 75: two studies to your knowledge. REF 1 also has that comparison.

Thank you for the suggestion, REF 1 has been added.

Line 346: these two formula are weird to present in the main text. Maybe put it into figure 2B.

We find it important to preserve the formula in the text as it is one of the main results and focal points of the manuscript. However, as requested, we now also added in into the figure for reader’s reference.

Line 517-520: the reference is a review – what is the actual piece of research showing this?

We have now added a reference to an original research article to support the statement.

Line 520-521: in the referenced study is actually just the smooth muscle cells co-culture that boosts myogenin – the tri-culture compared to this one is non-significant. Also, more myogenin expression ≠ better myogenesis.

Thank you for pointing this out. We adapted the statement and changed the tri-culture to co-culture of smooth muscle cells and bovine satellite cells.

Line 559: how can you conclude this based on the 90% DEGs being downregulated?

We now added a sentence in lines 643-645 to clarify the statement.

Line 590: RARRES1 genes has also been shown to be less expressed in bull calf samples vs dairy cow samples (bull calf showed better myotube formation)(REF 1).

Thank you for this insightful remark, we have added it into the discussion (lines 679-680).

Line 602: should it not be genomic stability?

Thank you, genetic stability is now changed to genomic stability.

REF 1: <https://doi.org/10.1016/j.foodres.2023.113217>

REF 2: [10.1016/j.devcel.2021.11.022](https://doi.org/10.1016/j.devcel.2021.11.022)

REF 3: <https://doi.org/10.3389/fnut.2023.1212196>

Reviewer #2 (Remarks to the Author):

Overall, the presented work by Olenic et al. includes a couple of interesting and potentially unexpected observations. The first is that there is not a major difference in performance between cells derived from fetal and adult cows. The second is that for at least a subset of myoblast cell cultures, proliferation beyond the hypothetical 'Hayflick limit' was observed. The general observation that primary cell cultures show reduced proliferation and differentiation over time is fairly well established at this point.

Overall, whilst these are interesting initial observations, in my opinion they are not yet a complete story that adds significantly enough to the cultivated meat field to warrant publication in a journal such as Communications Biology.

Comments:

The use of serum-containing medium for cell proliferation (and differentiation) is quite out-of-date for the field by this stage, and there are a number of formulations (Stout 2022, Kolkmann 2022, Messmer 2023) that could be tested (some of which the authors cite). It is known that serum contains a lot of highly pro-proliferative factors, and whether the continued proliferation would also be observed in serum-free medium is unclear. This study mirrors the Ding et al 2018 paper quite closely, but that article attempted some mechanistic explanations of satellite cell aging in vitro and included small molecule inhibitor testing that is not present here.

We thank the reviewer for bringing up the use of serum-free medium. As also mentioned in the first comment of reviewer #1, we previously tried different formulations of the serum-free media published in the cultured meat field, namely E8 (10.1007/s10616-019-00361-y), SFGM (10.3389/fnut.2023.1212196), Beefy-9 (10.1007/s10616-019-00361-y), and Beefy-R (10.1016/j.biomaterials.2023.122092). Unfortunately, we were not able to expand bovine myoblasts sufficiently in any of the tested formulations. In SFGM, Beefy-9 and Beefy-R our cells failed to reach confluency under our culture conditions (Response Letter Figure 1), therefore, any further analysis was impossible.

To provide more details on the selection of the expansion medium, we decided to include additional results (Supplementary Figure 2 and 3) about:

- Experiments where p38 MAPKi was added to the proliferation medium (PM vs PM+) to prevent premature differentiation during expansion
- Experiments where 50% of serum was replaced with a serum replacement Ultrosor
- Experiments with E8 medium based on the performance of this medium in Kolkmann et al., 2020 and its iterations to improve cell adhesion, proliferation and differentiation

However, we would like to emphasize that the goal of this paper was not to find optimal cell culture conditions or find serum-free alternatives. The context in which PM+ medium was selected for the further experiments has now been highlighted in Supplementary Figure 2 and lines 379-385.

The beta-galactosidase experiments presented in Figure 3 are very unconvincing, it's not clear to me whether the staining is even specific. Comparing only two passages, when the cells showed well over 50 cumulative population doublings, also seems very minimalist.

We acknowledge the limitations of the beta-galactosidase assay, as we described in the discussion (lines 607-615). Given the fast doubling rate of the myoblasts, it seems that senescence is not a major problem. However, we suggest to use a combination of Ki67 and γ H2A.X testing in the future.

The OMICs experiments, whilst nicely performed, are very descriptive, providing little mechanistic insight into why different behaviours emerge, because there is no way to distinguish between cause and effect in terms of cell aging and loss of myogenic potential. In my mind, a more interesting transcriptomic comparison would be between old vs young cells, to give insight into why some cell lines are able to continue proliferating and to potentially inform strategies for medium or cell development to improve future cultures. The title of the article claims that this study helps to “understand the mechanisms behind reduced myogenesis after bovine myoblast expansion”, but I don't really see how, beyond the immediate observation that better differentiating cultures show higher expression of myogenic genes.

We acknowledge the limitations of our study. The results of our omic analysis are quite broad and do not provide a definitive solution for reduced myogenesis. However, the data allow targeted hypothesis generation and we believe it lays ground for further research. Indeed, in combination with cited literature, we propose several concrete interventions, such as targeting HEY1, revising the way of culturing cells (shifting from fast-paced cultures to cultures with resting periods), manipulation of intracellular calcium, injection of oxytocin and targeting LIMK2. These are listed in discussion and summarized in the last paragraph. Each of these interventions potentially provides for another paper. To better reflect that we pinpoint potential mechanisms, rather than fully understanding them, we adapted the title to “Addressing hurdles in cultured meat by exploring reduced myogenesis after bovine myoblast expansion”.

No flow cytometry plots are presented in the main figures or supplementary data, making it impossible to ascertain the quality of these data and independently assess the authors claim that overgrowth of different cell types is not a factor in the continued proliferation.

We would kindly like to point out that Supplementary Figure 1A (referred to in the Materials and Methods) presents the gating strategy of CD56 marker expression. If individual scatterplots are required for each donor, we are happy to still add these later on as Supplementary data.

The doubling time below which cultures were excluded from subsequent experiments is set by rather arbitrary cut offs.

We understand the concern of the reviewer, but like to emphasize that once cultures reached a doubling time of more than 3 days, very limited proliferation was observed further on. This is indicated by a steep increase in the proliferation plots (Figure 1D and E).

The discussion is extremely lengthy (longer than the results section itself) and hence starts to stray towards speculation that goes quite far beyond the actual results presented in this study.

We understand the concern of the reviewer, however we believe that a thorough discussion is an asset for a high quality paper. Moreover, comparing our omic results with literature on

myogenesis provides ideas for further research to help solve the problem of reducing myogenesis. The paper currently adheres to all of the author's guidelines of the journal. Of course, in case the editor would want to impose further constraints on the length of the discussion, we are willing to comply.

Overall, the data is generally of a good quality (Figure 3 aside), but in my opinion this represents a promising set of observations to start off a project, rather than a complete work, and hence would need substantial further experiments to be considered for publication.

We agree that the manuscript provides a promising set of observations, which was also the goal of the paper – providing mechanistic insights for further research in the cultured meat field. In our opinion, exploring each strategy suggested would be too much for one publication given the diverse approaches mentioned. However, we do believe that with the additional data, valuable and unique information on bovine myoblast expansion and differentiation is provided.